# Using Pregnancy-Associated Glycoproteins (PAGs) to Improve Reproductive Management: From Dairy Cows to Other Dairy Livestock

**DOI:** 10.3390/ani12162033

**Published:** 2022-08-10

**Authors:** Olimpia Barbato, Laura Menchetti, Gabriele Brecchia, Vittoria Lucia Barile

**Affiliations:** 1Department of Veterinary Medicine, University of Perugia, Via San Costanzo 4, 06126 Perugia, Italy; 2Department of Veterinary Medicine, University of Milano, Via dell’Università 6, 26900 Lodi, Italy; 3Research Centre for Animal Production and Aquaculture, Consiglio per la Ricerca in Agricoltura e l’Analisi dell’Economia Agraria (CREA), Via Salaria 31, 00015 Monterotondo, Italy

**Keywords:** pregnancy-associated glycoproteins, bovine, sheep, goat, buffalo, dairy livestock, pregnancy, reproductive management

## Abstract

**Simple Summary:**

Pregnancy loss is a major cause of infertility in dairy animals, particularly in cattle, which affects the productivity and profitability of farms. Detecting these unsuccessful pregnancies could offer farmers the opportunity to reduce the economic damage caused by pregnancy loss. The determination of proteins secreted by the placenta and related to the presence of a viable conceptus called pregnancy-associated glycoproteins (PAGs) represents a diagnostic tool to identify pregnant or non-pregnant animals and to predict early pregnancy failures. This review describes the state of the art related to PAGs’ function, pregnancy profile, and use in reproductive management in bovine and other dairy livestock.

**Abstract:**

Pregnancy success represents a major issue for the economic income of cattle breeders. Early detection of pregnant and non-pregnant animals, as well as the prediction of early pregnancy failure, can influence farm management decisions. Several diagnostic tools for pregnancy are currently available. Among these, pregnancy-associated glycoproteins (PAGs) have been shown to be useful for identifying the presence of vital embryos and for pregnancy follow-up monitoring. This review presents an overview of the PAGs’ functions, their pregnancy trends, and their use as a tool to improve reproductive management in bovine and other dairy livestock, such as small ruminants and buffalos.

## 1. Introduction

In dairy production, reproductive performance and profitability are strongly associated. Declining fertility is a globally recognized problem that represents a major source of economic loss and culling in large ruminants [1,2,3]. Many factors contribute to the decline in reproductive efficiency. Among these, embryonic mortality represents the major cause of reproductive failure [4]. In cattle, following artificial insemination (AI), embryo loss can range from 40% to 65% by day 42 of gestation [5], leading to considerable economic losses spent on rebreeding animals, slower genetic progress, as well as significant losses in the number of possible calves [6]. Approximately 30% of all embryos and fetuses will not survive to birth. The introduction of intensive breeding systems and the selection of high-production animals have accentuated embryo mortality, either in bovine animals or in buffaloes, since high-production animals are much more at risk of embryo mortality than medium or low-production animals [7,8]. The estimated annual economic losses for farms due to embryonic mortality are reported to be $1.4 billion and $250 million in the United States of America and the United Kingdom, respectively, and $1.28 trillion worldwide [9].

There are management practices that can be implemented to reduce the economic damage caused by the presence of embryonic mortality. One strategy for improving reproductive performance aims to shorten the calving–conception interval by early pregnancy diagnosis to reinseminate non-pregnant animals as soon as possible [10]. Using ultrasonography as a method to detect pregnancy, the sensitivity of pregnancy diagnosis is 100% at the time of heartbeat detection, but the accuracy is limited before 28–30 days of gestation. Embryo death can be visibly diagnosed by an undetectable heartbeat; nevertheless, when an embryo with a heartbeat is viewed, there is no indication as to whether or not embryo mortality will occur. Therefore, ultrasonography is not predictive of embryonic mortality [11]. Thus, new technologies to identify embryonic mortality early after AI may play a key role in management strategies to improve the reproductive efficiency and profitability of dairy farms. An ideal early pregnancy test for dairy animals would fulfill the following criteria: high sensitivity (i.e., correctly identifying pregnant animals), high specificity (i.e., correctly identifying non-pregnant animals), be inexpensive to conduct, be simple to conduct under field conditions, and be able to determine early pregnancy failures. A test that summarizes these characteristics is that involving the determination of pregnancy-associated glycoproteins (PAGs) originating from mononucleate and binucleate cells of the embryonic trophoblast. Assays for detecting the PAG levels in maternal circulation have been developed and commercialized in cattle as in other livestock species [12].

This review provides an overview of PAGs’ characteristics and functions, their pregnancy trend, and their potential role in monitoring pregnancy in bovine and other dairy livestock.

## 2. PAGs’ Origins and Characteristics

The existence of present and detectable specific (PSP) or pregnancy-associated (PAG) proteins in maternal circulation was first described in 1982, when two proteins were isolated from the bovine placenta by immunoelectrophoresis: PSP-A and PSP-B [12]. PSP-A, with a molecular weight (MW) of 65–70 KDa and an isoelectric point (pI) of 4.6–4.8, has been identified as α-fetoprotein that is not strictly limited to pregnancy; PSP-B was found to be a specific gestation protein with a MW between 47 and 53 KDa and a pI of 4.0–4.4 [13,14]. About 10 years later, a pregnancy-associated protein with a MW of 67 KDa and four isoelectric forms (pI = 4.4, 4.6, 5.2, and 5.4), which differ from that of bovine placental tissue, was purified and discriminated on the basis of the percentage of sialic acid content [15]. The basic form, which has the highest immunoreactivity [16], was finally identified as bPAG-1 [17]. The similarity between bPSP-B and bPAG-1 was demonstrated by determining their nucleotide sequence, from which it was found that the cDNA of these two proteins is similar [18], but not identical. For bPAG, the name “associated” was preferred over “specific”, because it was also found in gonad extracts of non-pregnant males and females. PSP-60 is another pregnancy protein with a MW of 60 KDa, which can be considered similar to the other two. It was also measured during the cow’s gestation, and it was seen to increase from 20 or 27 days post-insemination [19] to 20 days pre-partum [20].

PAGs are a family of glycoproteins belonging to the sub-class of aspartic proteases [21,22], proteolytic enzymes with an acid pH; they are, in fact, characterized by the presence of aspartic acid residues surrounding the recognition sites and are similar to pepsin, renin, cathepsin D and E, and chymosin [23]. These proteins have a sequence that is 50% similar to that of pepsin [24], but the substitution of amino acids in the active site renders them enzymatically inactive [25]. Other authors successively showed that PAGs conserved features typically found in functionally aspartic proteins; therefore, some of them possess proteolytic activity [26,27].

PAGs are expressed in the outer epithelial layer (chorion/trophectoderm) of the placenta in eutherian species of the Cetartiodactyla order (even-toed ungulates) [28,29,30,31]. They are synthesized by the mononucleate and binucleate trophoblastic cells and secreted into the maternal blood [16,32].

From a phylogenetic point of view, PAGs can be categorized as “ancient” (originated about 87 million years ago), expressed in both mono- and binucleate trophoblastic cells (PAG-2 group), and “modern” (originated about 52 million years ago), expressed only in binucleate cells (PAG-1 group) [33]. Bovine PAG-2 coexists with Bovine PAG-1 in the trophectoderm [12,30,34]; while PAG-1 is produced in binucleate cells of both intercotiledonary and cotyledonary chorions, PAG-2 molecules are produced in both mononucleate and binucleate trophoblastic cells [32,35].

In ruminants, the PAGs gene family is particularly large and complex: cattle, sheep, and other pecoran mammals possess 100 or more PAG genes [29]. In bovine species, 22 PAG genes (boPAG-1 to boPAG-22) have been cloned and completely sequenced [21,30,36,37]. The number of identified PAG polypeptide precursors is lower in ovine (11 ovPAG) [29,36], caprine (12 caPAG) [37,38], cervid (10 cePAG) [31], and water buffalo species (wtPAG-1 [39]; wtPAG-2 to wtPAG-19) [12].

Not all PAGs are present in the same stage of gestation, as some appear earlier and others later. Green et al. [30] highlighted the existence of spatially and temporally distinct expression patterns during pregnancy. Some PAGs are completely expressed in the trophectoderm (oPAG-2, bPAG-2, bPAG-8, bPAG-10, and bPAG-11, where the numbers reflect the temporal order of discovery), while others are localized mainly in binuclear cells: among the latter, some (bPAG-1, bPAG-6, and bPAG-7) are present from mid- to late-pregnancy, while others (bPAG-4, bPAG-5, and bPAG-9) already appear at the 25th day, but are absent in the advanced stages [30]. Subsequent works in other species also confirmed a different trophoblastic distribution [26,31,32,37,38].

The different PAGs’ temporal expressions opened the way toward practical use and also toward speculation on their potential physiological role.

## 3. PAGs Functions

Given that the peculiar characteristic of PAGs is that their level constantly increases throughout the course of gestation, it is reasonable to believe that this is related to their biological function. In fact, PAGs were thought to be hormones binding to the surface receptors of maternal cells or proteins that sequestered or transported peptides; however, the discovery of the existence of different PAGs, each with its own structural specificity, does not seem to favor this theory [24]. Since PAGs and PSPBs were identified, several authors have searched for a relationship between these molecules or their concentration profile and local or systemic immunological functions. Telogu et al. [26,27] highlighted that some PAGs possess proteolytic activity. It is likely that proteolytically active PAGs present at the placenta–uterine interface could stimulate the synthesis of local growth factors [40,41]. Another possible function is that the PAGs’ presence at the placenta–uterine interface might play a role in adhesion by acting as bridging molecules [34].

According to Roberts et al. [24], PAGs could be able to bind and sequester peptides susceptible to recognition by the major histocompatibility complex (MHC) and to exert an immunomodulatory role at the maternal–fetal level; the local immunosuppression that takes place at the beginning of pregnancy is recognized to be necessary for the establishment and the maintenance of the maternal–fetal unit’s histocompatibility. In bovine animals, several works have correlated the high concentrations of PAGs with the decrease in the activity of polymorphonuclear neutrophils [42], suggesting that the trophoblast PAGs’ production, influencing the maternal immunological status, could be a mechanism by which the conceptus protects itself from rejection.

Austin et al. [43] attributed to PAGs a hormonal role in inducing the release of granulocyte chemotactic protein-2 (GCP-2), an alpha chemokine whose synthesis is induced by interferon-tau (IFN-tau) in early pregnancy. IFN-tau and PAGs would, therefore, share a common role in the stimulation of this chemokine, which appears to be involved in the onset of pregnancy in cattle.

In support of a possible luteotropic role of PAGs, the studies by Del Vecchio et al. [44] and Weems et al. [45] showed how these glycoproteins induce the release of prostaglandin (PG) E2 and progesterone from luteal cells and PGE2 from endometrial cells grown in vitro. In addition, bPAG-2 has been shown to be very similar to LH; in fact, it forms a bond with CL receptors, immunological kinship, and shows the same behavior in the course of purification as the hormone. For this reason, it was considered to represent one of the luteotropic factors of the placenta of ruminants [25].

Many authors have highlighted the relationship between PAGs/PSPB synthesis and fetal well-being [23,46,47]. The relationship between PAGs and fetal well-being is clearly understood if we consider that these glycoproteins are secreted by the mono- and binucleate cells of the trophectoderm, which migrate from the fetal to the uterine tissue and fuse with maternal uterine epithelial cells to form hybrid feto–maternal trinucleate cells, which are responsible for the release of glycoproteins in the maternal organism. This is an active process that presupposes the presence of healthy trophoblastic tissue and, therefore, of a healthy embryo. If this condition fails, the source of production of the proteins themselves is missing [23,46,47]. PAG-2, in particular, synthesized not only by the binucleate cells, but also by mononucleate cells of the trophectoderm, could result in an even earlier expression during the course of placenta formation. Furthermore, owing to this “active migration”, PAGs would play an important role in the remodeling of fetal membranes during the course of pregnancy [23].

## 4. PAG Trend during Gestation and Post-Partum

In cows, PAGs have been detected in the maternal blood as early as days 15 to 22 [14] or day 22 [48] after fecondation. The PAGs detection results are more accurate from days 28 to 30 onwards [49]. The levels of PAGs progressively increase in pregnant animals between the 6th and 35th weeks, and continue to rise more rapidly thereafter between the 35th and last weeks of gestation (Figure 1) [16]. Their concentrations then triple, more precisely between 20 and 10 days pre-partum [47], up to their maximum increase, which occurs about 5–10 days before calving [16].

After calving, the PAG concentrations begin to disappear from maternal blood to return to the baseline levels, but persist in an appreciable concentration for 80–100 days (Figure 1) [16,50,51]. The presence of PAGs in the plasma of cows during the early stages of the postpartum period may limit their use under field conditions. To diagnose pregnancy using PAG tests in the early stages of the post-partum period, the interval between calving and AI should be at least 60 days [50]. In this case, the residual PAGs in the maternal circulation during the postpartum period minimally interfere with a new subsequent pregnancy.

The slow disappearance of PAGs from maternal blood after calving is due to the presence of high concentrations at calving, and also to the long half-life of bovine PAGs, estimated to be 7.4–9 days in European cows [14,50,52] and around 9.2–10.1 days in African Azawak zebu cows [53].

The use of milk to determine the PAG levels has caught the interest of breeders, because milk sampling avoids the stressful effects of blood sampling, does not require special expertise, and is inexpensive. In cattle, the first study on the determination of the PAG concentrations in milk was conducted by Tainturier et al. [54] in the postpartum period. The evidence of PAG in milk suggested the possibility of utilizing milk to also detect PAG during early pregnancy. Few studies have been reported in the literature about the PAG concentration in milk throughout pregnancy in bovine animals. There have been only a few assays described that quantify bovine PAGs in milk [55,56,57,58]. Metelo et al. [56] found that the concentrations of PAGs in milk appear to be 10 to 50 times lower than those of the plasma, but their profile during the course of pregnancy is comparable. In the post-partum period, however, the decline in the PAGs’ concentrations is faster in the milk than in the blood [57]. It has been shown that, within 6 days of calving, the PAGs’ concentrations decrease by 50% in plasma and by 95% in milk [54].

Recently, Krebs et al. [59], using ELISA newly developed in their lab, reported that the amount of PAG found in milk was 1.5% of the amount present in serum. The milk test identifies pregnancy at day 40 post-insemination, with a threshold value of 0.0165 ng/mL.

Low concentrations of PAG in milk samples is still a problem for pregnancy diagnosis and, although over the last few years an increasing sensitivity of the PAG method was noted, the test needs to be implemented [57,58].

In our opinion, the determination of PAGs in milk, even though it is a stress-free and non-invasive method, is not yet feasible as an early pregnancy test; therefore, it is not effective in management strategies to improve the reproductive efficiency and profitability of dairy farms. As reported by LeBlanc [60], the utility of the milk pregnancy test may be attributable to the convenience of confirmation of pregnancy in cows previously diagnosed pregnant, without handling the animals. In any case, it has been suggested to check the cows indicated as not pregnant by the milk PAG test by routine examination methods (i.e., rectal palpation or ultrasonography) to confirm the non-pregnant status before starting a new cycle of synchronization and re-insemination.

## 5. PAGs and Fetal Well-Being

The use of PAGs is not only beneficial for advancing a pregnancy diagnosis, but also for obtaining information on embryonic and/or fetal well-being. The PAGs’ concentrations, for example, begin to decrease after 1–3 days of experimental infection with *Actinomyces pyogenes*, which is an abortive agent, reaching values below 0.6 ng/mL in about 30 days [61]. Even in pregnant goats following *Toxoplasma gondii* and *Listeria monocytogenes* infection, the PAG profile agrees with the action kinetics of the abortion agent: in Listeria-infected goats, the proteins’ concentrations drop drastically the day after inoculation and 9 days before abortion; in those infected with Toxoplasma, the PAGs’ concentrations decrease until the day of expulsion of the fetus [62]. Lower plasma PAG concentrations throughout gestation have been reported in cows naturally infected by *Neospora caninum*, which is an abortigen agent, with respect to their seronegative peers [63], while in cows suffering Neospora-associated abortion, the PAG concentrations were found to fall drastically [64]. The mechanism for this decline seems to be linked to a reduction in the activity of trophoblast cells. Therefore, PAGs can be considered as a marker of placental function and a direct tool to monitor the vitality of the conceptus.

The possibility that the PAG profile could reveal embryo loss was described for the first time by Szenci et al. [51]. They reported that the plasma concentration of PAGs decreased in cows experiencing embryonic/fetal mortality as assumed by ultrasonography. They found that the changes in concentrations were similar to those reported after experimental infection with *Actinomyces pyogenes* by Semanbo et al. [61].

Polher et al. [65], in a trial that examined the relationship between the day-31 circulating concentrations of PAGs and embryonic survivability, reported that the concentrations of PAGs on day 31 of gestation may provide a good marker for predicting embryonic mortality between days 31 and 59, suggesting that the analysis could help predict which cows will undergo embryo mortality.

Recently, many authors have shown the relationship between the PAG level and embryonic losses [66,67,68,69,70], showing that use of PAGs’ concentrations can help, in addition to pregnancy diagnosis, in new investigations of embryonic or fetal mortalities (Figure 2).

## 6. Other Livestock Species

### 6.1. Sheep and Goats

In small ruminant farms, profitability is obtained by increasing the difference between the selling price of milk and its production cost. The sale price is generally determined and influenced by entities outside the company and the bargaining margin is very limited, if not absent. In some cases, cheese can be produced to increase the surplus value of the milk. It is, therefore, important for the farmer to reduce the cost of milk production, without neglecting its quality, while maximizing animal welfare, zootechnical performance, and reproductive parameters, which are the bases of the sustainability of sheep farms [71].

In particular, the reproductive management in the breeding of small ruminants must take into account the economic conditions and the availability of low-cost food resources in the area. Early pregnancy diagnosis is an indispensable tool in farm management—it allows to correctly plan lambing management and to properly formulate rations for the energy requirements of gestating or lactating animals at the same time.

#### 6.1.1. Sheep

The pregnancy-related proteins identified in sheep were named SBU-3 [72], PSPB [73], and PAG [20]. Ovine PAGs are glycoproteins with a MW ranging from 43 to 70 kDa and a pI ranging from 4 to 6.8 [17,74].

In ovine placentas, 11 types of cDNA coding for distinct PAGs (ovPAG-1 to ovPAG-11) were identified at different gestational periods, thus confirming the multiplicity and temporal expression of PAG molecules in ruminant placentas [12,32,36]. These molecules are present in maternal plasma at concentrations detectable as early as 18–22 days after conception [75,76,77,78,79,80,81,82,83,84]. In this species, the PAGs’ concentrations differ throughout pregnancy according to the breed, fetal number, sex, and birth weight [85,86,87,88].

In Lacaune and Sarda sheep, the PAGs’ concentrations increase during the gestation period up to 60 days, and then decrease until 120 days; thereafter, there is a significant rise, which peaks at parturition (Figure 3) [89]. This is a similar trend to that described for various sheep breeds, i.e., Berrichon or Churra [76,79,86,90], but not for Merinos ewes, whose PAG levels initially increase and then drop to baseline concentrations around mid-pregnancy. Contrary to what happens in cows, the PAGs’ profile during gestation is biphasic, with two maximum levels; a pre-calving peak is not observed, and the concentration reaches basal levels already after four weeks post-partum. The origin of this difference is thought to be due to the structure of the placenta—in fact, a decrease in the number of binucleated cells is observed in sheep [91].

In the post-partum period, the PAGs’ concentrations progressively decrease, reaching a value of 0 ng/mL at 28 days after lambing (Figure 3) [89]. A similar decrease during the first month post-partum was reported by Ranilla et al. [76,86]. Unlike in cows, the rapid PAG disappearance in ewes does not require the use of a cut-off limit in postpartum animals as a means for detecting a new pregnancy. The half-life was calculated to be between 4.5 [50] and 6.9 days from calving [89], fewer than the 9 days recorded in cows.

Assaying PAGs in milk could be an alternative for pregnancy diagnosis in this species. It has been reported that the milk assay can provide an accurate diagnosis from day 32 of gestation onward. Regarding the effect of the litter size, no clear relationship was observed with the PAGs’ concentrations in milk [81].

#### 6.1.2. Goats

Three different PAG molecules were isolated and partially characterized from goat placentas, with MWs of 55, 59, and 62, showing several isoforms [92]. In caprine placentas, Garbayo et al. [38] described 11 types of cDNA coding for PAG molecules (caPAG-1 to caPAG-11). Nine of those molecules, named the caPAG-1 group, are expressed in trophoblast binucleate cells, while boPAG-2 and ovPAG-2, caPAG-2 were found to be expressed throughout the trophectoderm during early pregnancy (Days 18 and 19). As described in bovine species, caPAG-2 is of a more ancient origin than the caPAG-1 group.

In order to develop an accurate RIA system, two semi-purified preparations of caPAG allowed the production of two antisera (AS#706 and AS#708), by which it was possible to identify pregnant animals as early as 21 days after insemination [93]. During gestation, the PAGs’ concentrations reach the maximum levels in week 8, decrease between weeks 12 and 14, and remain relatively constant until parturition. After parturition, the concentrations decrease rapidly, reaching lower levels at the fourth week post-partum [94,95] (Figure 4). PAGs were eliminated by a two-step log-linear decline, and the estimated postpartum half-lives were 3.6 and 7.5 days in the initial fast and terminal slow phases [49].

The plasma PAG concentration was reported to be similar in different breeds, such as Moxoto goat [96], Alpine goat [97], Carpathian goat [98], Boer goat [99], and Barbari goat [100], determined through heterologous RIA or ELISA techniques.

As observed in cattle, both the number and genotypes of fetuses can influence the PAGs’ concentrations. The PAGs’ levels in maternal circulation are higher in twin-bearing goats than in goats bearing one fetus [95,96], and are also higher (about ten times) in interspecific pregnancies (Spanish ibex embryos transferred to domestic goats) when compared with that found in normal intraspecific gestation [101].

The consecutive measurement of PAGs in goats also allows for the determination of trophoblastic activity disorder leading to the death of a fetus [94,97,102].

Although the use of specific antisera, as reported before, allows discrimination between pregnant and non-pregnant goats as early as 21 days after breeding, its use in farm conditions is recommended after day 24 post-AI [103].

PAGs are also detectable from day 32 after conception in the milk of pregnant goats, while they are almost absent in non-pregnant goats [104]. A very recent study using ELISA [105] reported day 26 post-mating as the first time-point for significantly higher milk PAGs’ concentrations and day 37 post-mating as the best suitable time point to detect pregnancy in goats. Doğan and Köse [106], utilizing the visual PAG-ELISA test, reported that it was able to diagnose pregnancy in goats starting from the 28th day after breeding, but its reliability still needs to be verified, since visual results are subjective.

### 6.2. Buffalo

The worldwide buffalo population is increasing continuously, and has been estimated to be 204 million head with an increase of 5% in the last 10 years (FAOSTAT, 2019). Ninety-seven percent of the buffalo population is concentrated in Asia, mainly in India and Pakistan, due to dairy breed selection and the enhancement of the milk market. There is also an increasing trend in Europe and America due to the growing number of dairy animals, related to the world market’s high demand for milk, cheese, and processed products [107].

Reproductive efficiency in buffalo is affected by seasonality, particularly during the spring–summer season, which corresponds to the low breeding period for this species [108,109]. Buffaloes that conceive during the daylight-lengthening period (i.e., spring-summer) showed a higher incidence of embryo loss (20 to 40%) than those that conceived during decreasing daylight length (7%) [110,111].

The RIA-706 system, which uses antisera raised against caprine PAGs, was the first to be adopted for detecting PAG molecules in buffaloes. With this system, pregnancy was detected in the time window from days 31–35 after mating [112]. El-Battawy et al. [113], using the same RIA system, observed great differences in the PAG concentrations between pregnant and non-pregnant buffaloes from day 28 onward.

Subsequently, the isolation and purification of PAGs from buffalo placenta allowed the development of a specific RIA system for buffaloes (RIA 860) [114,115,116], improving the accuracy of detecting pregnant buffaloes on day 25, reaching 99% on day 28 of gestation. Using this system, the buffalo plasma PAG profiles were described during pregnancy and the post-partum period (Figure 5 and Figure 6) [116].

During pregnancy, the PAG concentration increased up to day 105 and then remained constant until parturition, different from bovine animals, for which, as described before, an exponential increase was observed until parturition. In the post-partum period, the PAG concentrations decrease rapidly, reaching minimum values (<1 ng/mL) on day 30. Unlike bovine animals, the rapid PAG disappearance in buffaloes does not require a cut-off limit in post-partum animals as a means for detecting a new pregnancy, as there is a voluntary waiting period of at least 50 days. The half-life, found to be 8.5 days, proved to be shorter in buffaloes than in bovine animals [116].

In a subsequent study, the RIA-706 system showed greater sensitivity and accuracy at both 23 and 25 days of pregnancy with respect to the RIA 860, proving to be more efficient for early pregnancy diagnosis [117].

PAG assays, as demonstrated by different authors [112,117,118,119], can be reliable biomarkers for early pregnancy detection and embryo mortality in buffaloes (Figure 7).

Recently, Barile et al. [11] showed that PAGs permitted the discrimination between buffalo that experienced embryonic mortality and those that maintained pregnancy starting from 25 days of gestation, defining the optimal cut-off value for predicting mortality at day 25 and 28 post-AI.

Despite the fact that there have been various works on bovine animals [120,121,122,123] in which PAGs were measured using also antibodies raised against PAG-2, to our knowledge, only two works exist in which this method was reported for buffalo species [124,125]. The results of these works showed that the threshold value of 1.0 ng/mL was not reached before day 40 post-AI, although significant differences between pregnant and non-pregnant buffaloes were found starting from Day 28 post-AI.

To better understand the role that PAGs play during pregnancy in buffaloes, studies to assess PAG-2 mRNA expression in the maternal subset of blood leukocytes at the peri-implantation period were conducted. The studies showed that PAG-2 mRNA can be detected in peripheral maternal blood cells earlier than PAG-1 circulating placental proteins, and the quantification of PAG-2 mRNA could be used to differentiate pregnant and non-pregnant buffaloes, starting from the second week post-AI [124,125]. At the moment, PAG-2 mRNA could be a useful marker for studies on early pregnancy and embryonic mortality in buffaloes as in other ruminant species, but not for use in the field because of the cost and the procedure of the single analysis.

## 7. Conclusions

The improvement of the reproductive efficiency of dairy cows as in other dairy livestock is a preeminent goal within dairy farm management.

In veterinary practice, PAG measurement is a feasible method of early pregnancy diagnosis, pregnancy confirmation, and follow-up determination of embryo vitality. The first aspect can help breeders in the management of reproduction, allowing early resynchronization and rebreeding of the non-pregnant animals, aiming to shorten the calving–conception interval.

The identification of subjects at risk of embryonic mortality can influence management decisions for the recovery of those animals that would experience pregnancy loss, i.e., through pharmacologic intervention strategies for the maintenance of pregnancy.

Finally, the quantification of PAG mRNA expression could be a useful tool to better understand the mechanisms involved in early embryo development and create new strategies for therapeutic intervention to prevent early pregnancy loss.

## Figures and Tables

**Figure 1 animals-12-02033-f001:**
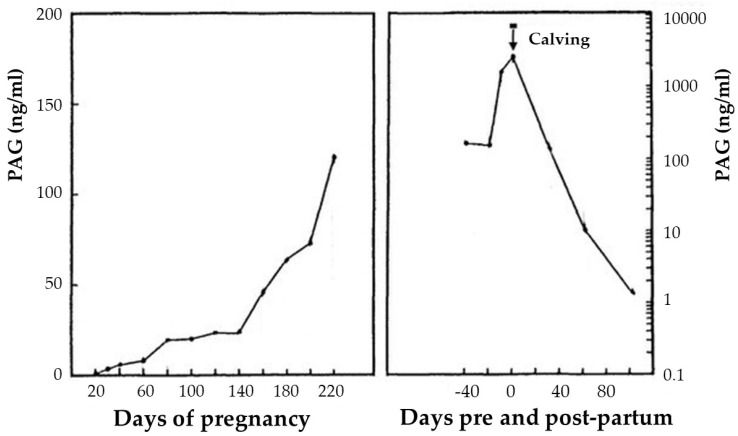
Pregnancy-associated glycoprotein (PAG) plasma profile of a cow during pregnancy and the post-partum period. Adapted from Zoli et al. [16].

**Figure 2 animals-12-02033-f002:**
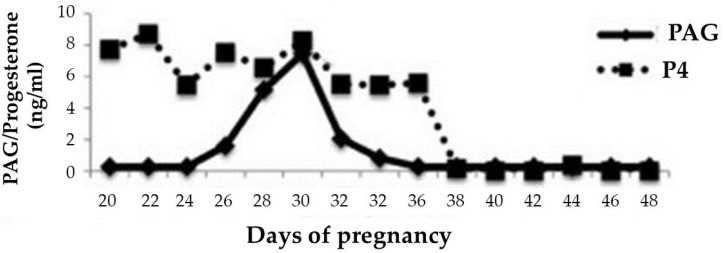
Pregnancy-associated glycoproteins (PAGs) and progesterone (P4) in a cow that experienced embryonic mortality between days 30 and 40 of gestation. Adapted from Pohler et al. [69].

**Figure 3 animals-12-02033-f003:**
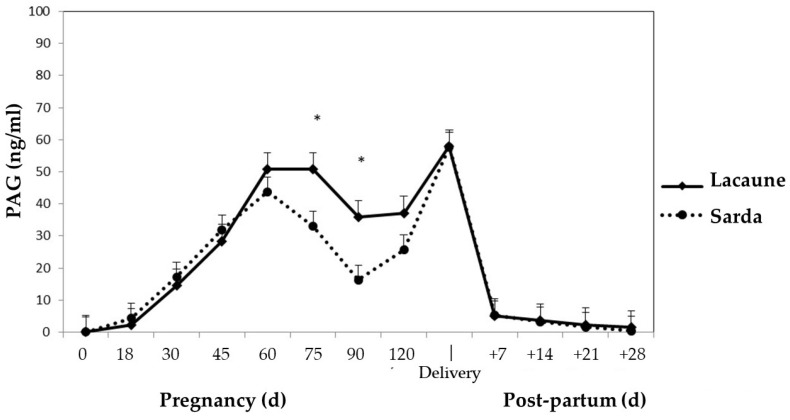
Pregnancy-associated glycoprotein (PAG) plasma profile in Lacaune and Sarda ewes during pregnancy and the post-partum period. * *p* < 0.001. Adapted from De Carolis et al. [89].

**Figure 4 animals-12-02033-f004:**
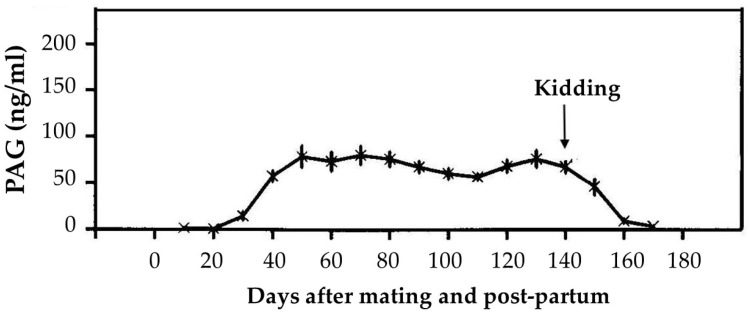
Pregnancy-associated glycoprotein (PAG) plasma profile during pregnancy and post-partum in goats. Adapted from Zarrouk et al. [94].

**Figure 5 animals-12-02033-f005:**
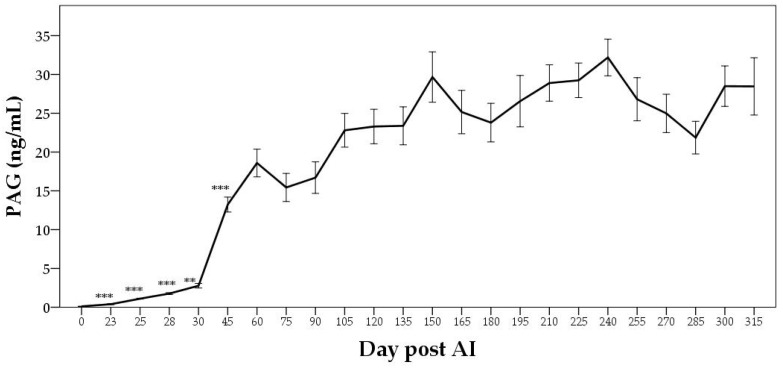
Pregnancy-associated glycoprotein (PAG) plasma profile during pregnancy in buffalo cows. *** *p* < 0.001, ** *p* < 0.01 versus previous day. Adapted from Barbato et al. [116].

**Figure 6 animals-12-02033-f006:**
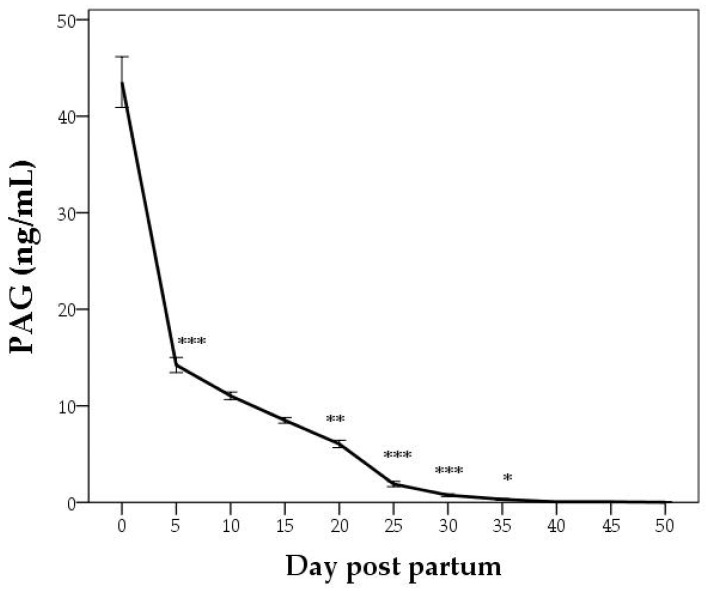
Pregnancy-associated glycoprotein (PAG) plasma profile in buffalo cows during the post-partum period. *** *p* < 0.001, ** *p* < 0.01, * *p* < 0.05 versus previous day. Adapted from Barbato et al. [116].

**Figure 7 animals-12-02033-f007:**
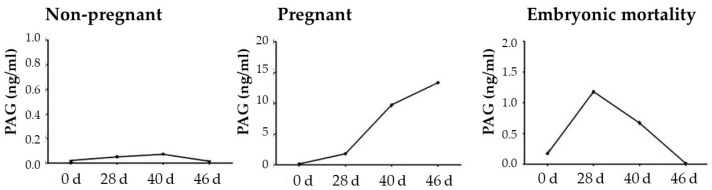
Pregnancy-associated glycoprotein (PAG) plasma concentrations in non-pregnant and pregnant buffalo cows, and those that experienced embryonic mortality. Adapted from Barbato et al. [119].

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
