# Peer review of "Using Pregnancy-Associated Glycoproteins (PAGs) to Improve Reproductive Management: From Dairy Cows to Other Dairy Livestock"

_animals, 2022, doi:10.3390/ani12162033_

Round 1

Reviewer 1 Report

Dear author

Bellow are the comments:

P2L62: A test which is summarizing instead of A test summarizes

P2L63: that of determination instead of that for the determination

P2L68: as indicators of pregnancy instead of as indicators or monitors of pregnancy

P2 L85: MW instead of PM

P2L86: What you mean by It was also dosed during the cow's gestation? May be yoy want to say it was determined?

P3L101: can be grouped as “old” instead of can be grouped as “ancient”

P4L178: The detection results are more accurate instead of The detection results more accurate

P6L225: as an early pregnancy test instead of as a early pregnancy test

P6L229-232: English needs to be improved in the bellow sentence. Moreover, the recommended action for cows with a milk PAGs test result of not pregnant is veterinary reexamination of the animal to confirm the presence or absence of a viable fetus before reinsemination or administration of prostaglandin.

P6L251: presented embryonic/fetal mortality instead of experienced embryonic/fetal mortality

P7L264: presented embryonic/fetal mortality instead of experienced embryonic/fetal mortality

P10L372: buffalo population instead of world buffalo population

P10L373: In Asia 97% of buffalo population is concentrated, mainly in India and Pakistan, instead of In Asia is concentrated 97% world’s buffalo population, mainly in India and Pakistan,

P10L392: from days 31-35 after mating instead of from days 31e 35 after mating

P10L396-397: allowed the development of a specific RIA system for buffalo instead of allowed for the development of a specific RIA system for buffalo

P11 L418: In another study instead of In a following study

P12L444: To understand better the role that PAGs play instead of To better understand the role that PAGs play

Reviewer 2 Report

This review is a good state of art of PAGs in ruminants. It would have been interesting to compare advantages and disadvantages of PAG evaluation and echography.  Personnaly, I am not convinced that PAG is an usefull toll to improve reproduction management in the different species.  In the field, echography is quite more often used. So the title could be changed.

A table comparing the concentrations of PAGs during gestation and postpartum in the different species may be advantageous to replace the figures presented.

More specifically : 

L35 you an add also milk production

L62 echography (b-mode and doppler mode) meet also some of such caracteristics. Could you add some comments ?

L87 post insemination could be better than postestrus

L109 what means pecoran animals ?

L233 what happen after an injection of PGF2a and the decrease of progesterone

L235 vitality could be better to use than well-being

L236 namely actually Trueperella pyogenes

L283 What means SBU ?

L296 What ares the hypothesis to explain such differences between sheep breeds ?

L392 31 to 35

Round 2

Reviewer 1 Report

Dear author,

 On the peer-review 2 you have to follow the bellow comments:

 P6 L236-240: The sentence has no sense. You have to re-write the sentence and improve the language
